# Event-ID: Intrinsic Decomposition Using an Event Camera

## ABSTRACT

Reconstructing 3D scenes from multi-view images is challenging, especially in adverse conditions. We propose a novel event-based intrinsic decomposition framework that leverages events and images for stable decomposition under extreme scenarios. Our method is based on two observations: event cameras maintain good imaging quality, and events from different viewpoints exhibit similarity in diffuse regions while varying in specular regions. We establish an event-based reflectance model and introduce an event-based warping method to extract specular clues. Our two-part framework constructs a radiance field and decomposes the scene into normal, material, and lighting. Experimental results demonstrate superior performance compared to state-of-the-art methods. Our contributions include an event-based reflectance model, event warping-based consistency learning, and a framework for event-based intrinsic decomposition.

## CCS CONCEPTS

• **Do Not Use This Code → Generate the Correct Terms for Your Paper**; *Generate the Correct Terms for Your Paper*; Generate the Correct Terms for Your Paper; Generate the Correct Terms for Your Paper.

## KEYWORDS

Do, Not, Us, This, Code, Put, the, Correct, Terms, for, Your, Paper

## 1 INTRODUCTION

Intrinsic decomposition from multi-view images is a fundamental task, that enables a range of downstream applications including view synthesis [5, 7, 62, 71], relighting [2, 5, 7, 21, 24, 39, 62, 73], object insertion [4, 19, 73], and digital heritage preservation [58]. Yet, present methodologies [5, 6, 32, 41, 51, 62, 75, 78, 83, 84] presuppose that the input images are sharp and well-exposed.

During the execution of downstream tasks, practitioners encounter various and demanding environments. These may involve scenarios such as reconstructing murals [64] in low-light caves, capturing scenes affected by severe camera shake-induced blur, or dealing with a high dynamic range that leads to over-exposed images. Under these conditions, the compromised image quality impedes the effectiveness of conventional methods, which struggle to discern precise cues for intrinsic decomposition, such as detailed gradient information and color distribution.

*ACM MM, 2024, Melbourne, Australia*
© 2024 Copyright held by the owner/author(s). Publication rights licensed to ACM.
ACM ISBN 978-x-xxxx-xxxx-x/YY/MM
https://doi.org/10.1145/nnnnnnn.nnnnnnn

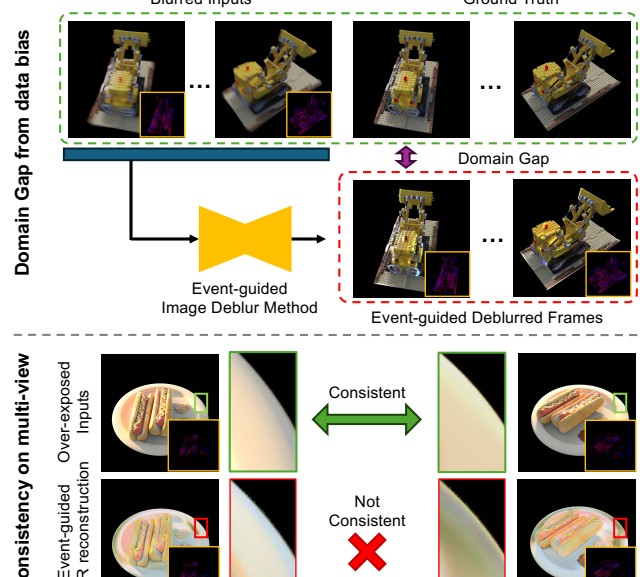

**Figure 1: Demonstration of two primary challenges posed by event-guided techniques. (Top) The domain gap between predictions from event-guided image deblurring and ground truth. While existing methods [69] enhance clarity in blurred inputs, they often introduce significant color shifts that can impact intrinsic decomposition results. (Bottom) Multi-view inconsistency arises from frame-by-frame event-guided HDR reconstruction [74]. The method neglects 3D consistency during HDR recovery, focusing solely on time-related event information. This lack of 3D consistency increase the ill-posedness of intrinsic decomposition and might severely compromise performance.**

Event cameras [16], with their high dynamic range and high temporal resolution capabilities, outperform traditional cameras in maintaining quality imaging under adverse conditions. The growing ubiquity of hybrid camera technology [10, 68], which integrates both events and images, paves the way for leveraging event information to substantially improve image quality. However, using event-guided methods [33, 35, 74] to enhance the input images does not yield satisfactory results (shown in Fig. 1). Firstly, most of these methods are data-driven, and when the test data has a domain gap with the training data, it can easily affect the results. Secondly, these methods typically process images on a frame-by-frame basis, which results in a loss of 3D consistency across multi-views [77].

To achieve this, we exploit the two primary attributes of events: 1) Event camera still maintains good imaging quality in these extreme scenarios[1]; 2) Events generated from different viewpoints

---

[1]Event camera is capable of capturing color information [57].

are relatively similar in diffuse regions, while in specular regions, they vary with changes in viewpoint. Based on the first point, we established an event-based reflectance model that captures the relationship between geometry, materials, lighting, and events in a 3D scene. Besides, with the second point, we introduce an event-based warping method that extracts specular-related clues from events to mitigate the impact of specular reflection on the intrinsic decomposition task.

To the end, based on the proposed reflectance model and specular-related clues from events, we propose an **Event**-based **I**ntrinsic **D**ecomposition method (**Event-ID**) taking the inputs of both events and images. In specific, we employ four separate Multilayer Perceptrons (MLPs) to represent the geometry, the surface normals, the material properties of the surface, and the scene lighting. Besides, we additionally modeling the specular variation on the changing of events to enable the specular-aware method, for recovering the challenge geometry and base color information on over-saturated specular region. Our method achieves state-of-the-art results under low-light conditions, overexposure, and when the input image is blurry. Compared to other methods, our approach can perform stable intrinsic decomposition even under extreme conditions. The main contributions of this paper can be summarized as follows:

(1) We build an event-based reflectance model that establishes the relationship between the geometry, materials, lighting, and events within a 3d scene.
(2) Observing the multi-view consistency of events, we extract specular-related clues from events and apply these clues to assist in intrinsic decomposition.
(3) Leveraging the event-based reflectance model and specular-related clues derived from events, we propose a framework for intrinsic decomposition using events, which enables relighting under extreme conditions.

## 2 RELATED WORKS

***Intrinsic decomposition from multi-view images.*** There are several works [4, 26, 37, 43, 52, 59, 80] on extracting scene geometry, materials, and lighting from multi-view images. Here, we focus on introducing the efforts related to neural representation [5, 6, 32, 41, 51, 62, 75, 78, 83, 84]. NeRFactor [83] recovers an object's shape and reflectance from multi-view images under a single unknown lighting condition. TensoIR [32] model secondary shading effects (like shadows and indirect lighting). PhySG [81] and NeILF++ [78] obtain surface normal by optimizing the Signed Distance Field (SDF). Building upon the foundation of NeILF [75], NeILF++ [78] incorporates inter-object reflections, yielding more accurate geometric information. GS-IR [41] employs 3D Gaussian Splatting for the estimation of scene geometry, surface material, and environmental illumination.

These methods typically require input images to be clear and well-exposed, resulting in subpar performance in scenarios where the image quality is compromised. Our approach capitalizes on the high temporal resolution and high dynamic range of events, ensuring stable performance even in challenging lighting conditions.

***Warping-based Consistency Learning.*** Warping-based consistency learning is commonly employed in multi-view stereo [65, 70, 72, 79], as well as in neural implicit surface learning [13, 15, 20],

for the purpose of 3D reconstruction. It harnesses the power of inter-image correspondences through differentiable warping operations to enhance the reconstruction process. NeuralWarp [13] applies to warp to points sampled along a ray towards source images for RGB value extraction. Geo-Neus [15] warps grayscale patches around the anticipated surface points to adjacent images, ensuring multi-view geometric consistency. Ref-NeuS [20] proposes leveraging an anomaly detection framework to compute an explicit reflection score, utilizing multi-view contextual information to accurately identify and localize reflective surfaces within the scene. When the imaging quality of the image degrades, the performance of these methods tends to diminish.

Current warping methods based on events are predominantly used for optical flow estimation [17, 18, 61, 85], motion estimation [1, 11, 34, 85], and event alignment [18, 23, 63]. These methods focus on 2D warping and alignment, without addressing the alignment of 3D scenes. Moreover, these warping techniques are grounded in contrast maximization approaches [18, 63], which all rely on the assumption of photometric consistency and fail to consider the specularities present within the events. In contrast, our method exploits the inconsistency of events between diffuse and specular regions to extract specular-related information from 3D scenes. Compared to image-based methods, my approach remains robust and yields stable results even in scenarios with poor image quality.

***Events for geometry, lighting, and material estimation.*** Event cameras, characterized by their high temporal resolution and high dynamic range, have been primarily utilized in previous research for depth estimation within scenes [18, 27, 48, 53, 85]. Additionally, there are efforts focused on reconstructing the three-dimensional shape of objects [3]. [22, 49] takes advantage of the high temporal resolution of event cameras by combining polarization information with events to estimate shape. Within the field of lighting estimation using event cameras, [12] exploits the high temporal resolution characteristic of events to capture the brightness changes of objects at the instant the light is switched on, in order to estimate the distance of the light source. To date, there has not been work on utilizing event cameras to estimate material properties.

To the best of our knowledge, we are the first to use event cameras for the estimation of geometry, materials, and lighting simultaneously.

***Events for neural radiance fields.*** Recent research [31, 36, 42, 44, 54, 57] has sought to integrate events with Neural Radiance Fields (NeRF) to improve neural radiance fields reconstruction. Given that event cameras only capture changes in intensity rather than absolute values, methods [31, 42, 57] that exclusively depend on events as inputs require the manual assignment of a background intensity value. This approach could potentially lead to inaccuracies in the intensity of the reconstructed scene. In response to this issue, other techniques [36, 44, 54] employ both images and events as inputs.

Although existing research has primarily focused on extracting radiance and geometric details from 3D scenes, they largely overlook the reconstruction of material properties and lighting

conditions. Our method stands out by reconstructing the geometry from events while concurrently capturing detailed material attributes and lighting nuances.

## 3 MODELING

In this section, we present our event-based intrinsic decomposition framework. Section 3.1 describes the events-based reflectance model, which captures the relationship between events and the geometry, materials, and lighting in a 3D scene. In Section 3.2, we explain how to extract specular-related information from events and how this information can be utilized in the intrinsic decomposition task. Finally, Section 3.3 provides a detailed description of our event-based intrinsic decomposition framework.

### 3.1 Events-based Reflectance Model

**Event generation model.** An event $e = (\mathbf{u}, t, p)$ at pixel position $\mathbf{u} = (u, v)$ and time $t$ with polarity $p \in \{-1, +1\}$ is generated when the logarithmic change of brightness $I$ since the last event at the pixel $\mathbf{x}$ and time $t - \Delta t$ exceeds a threshold $\Theta$ ($\Theta > 0$) and the event $e$ can be represented [55] as:

$$e(\mathbf{u}, t) = \left\lfloor \frac{\ln(I(\mathbf{u}, t)) - \ln(I(\mathbf{u}, t - \Delta t))}{\Theta} \right\rfloor. \tag{1}$$

**Event-based reflectance model.** From Equation (1), we can find that an event represents a change in brightness caused by the movement of the viewpoint in world space[2] within time $\Delta t$. As shown in Figure 2, We assume that at time $t$, an event camera captures the brightness of point $\mathbf{x}_t$ in the 3D scene at pixel $\mathbf{u}$, and at time $t - \Delta t$, the brightness at pixel $\mathbf{u}$ (image space) corresponds to the brightness of point $\mathbf{x}_{t-\Delta t}$ in the 3D scene (world space). The event-based reflectance model can be described as:

$$e(\mathbf{u}, t) = \left\lfloor \frac{\ln(L(\omega_o^t, \mathbf{x}_t)) - \ln(L(\omega_o^{t-\Delta t}, \mathbf{x}_{t-\Delta t}))}{\Theta} \right\rfloor, \tag{2}$$

$$L(\omega_o^t, \mathbf{x}) = \int_\Omega f(\omega_o^t, \omega_i, \mathbf{x}) L_i(\omega_i, \mathbf{x})(\omega_i \cdot \mathbf{n}) d\omega_i, \tag{3}$$

where $\omega_o^t$ is the viewing direction of the outgoing light at time $t$, $\mathbf{n}$ is the surface normal, $L_i$ is the incident light from direction $\omega_i$, and $f$ is the Bidirectional Reflectance Distribution Function [46] (BRDF) properties of the surface point $\mathbf{x}$.

For traditional images, there is a one-to-one correspondence between points in image space and points in world space. This means a pixel value represents the result of a physically-based rendering [56] process at a specific location in the 3D scene, taking into account the geometry, material, and lighting at that position. However, for event cameras, since an event describes a change in brightness, a position in image space corresponds to two points in world space. That is, a single event value is related to the physically-based rendering results of two different locations in the 3D scene, influenced by their respective geometry, material, and lighting conditions.

---

[2]In a static scene, this change in brightness is solely caused by the camera's motion.

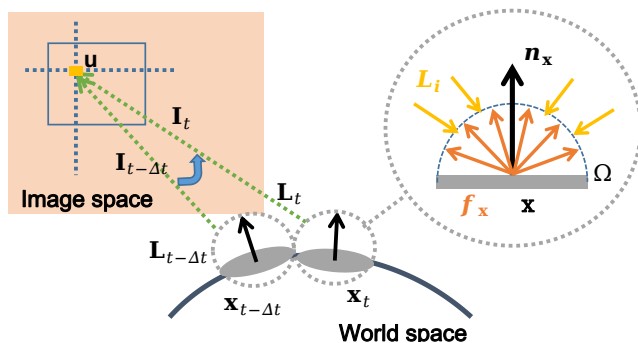

**Figure 2: Illustration of events-based reflectance model. In image space, the generation of an event $e(\mathbf{u}, t)$ represents a change in brightness from position $\mathbf{x}_{t-\theta t}$ to position $\mathbf{x}_t$ in world space. The color at a position $\mathbf{x}$ in world space is determined by the surface normal $\mathbf{n}_x$, the BRDF properties $f_x$, and the incident light $L_i$ at that position $\mathbf{x}$.**

### 3.2 Event Clues for Intrinsic Decomposition

In 3D scenes, the diffuse component of objects exhibits the property of photometric consistency across different viewpoints, while the specular regions are view-dependent [29, 45, 86]. By leveraging this characteristic, it is possible to separate the diffuse and specular regions, which facilitates intrinsic decomposition [14, 38, 76].

Image-based methods often employ patch warping techniques to project image patches from different viewpoints onto the same viewpoint [13, 15, 20]. By computing the similarity between these warped patches, it is possible to determine whether a specular region exists at that particular viewpoint. However, when the quality of the images deteriorates (such as underexposure, overexposure, or blur), this warping approach [13, 15, 20] becomes ineffective. The degradation in image quality can lead to inaccuracies in the patch warping process, making it challenging to reliably compare the warped patches and identify specular regions. Consequently, the effectiveness of these image-based methods diminishes as the quality of the input images decreases (shown in Figure 3).

Compared to traditional cameras, event cameras offer unique advantages such as high dynamic range and high temporal resolution. These characteristics enable event cameras to maintain good image quality even in extreme lighting conditions and rapid motion scenarios where traditional cameras may struggle. Therefore, in extreme situations where traditional image-based methods may fail due to underexposure, overexposure, or motion blur, events provide a robust alternative for acquiring reliable specular clues (shown in Figure 3).

**Event-based warping.** As described in Section 3.1, for images, the brightness of each pixel corresponds one-to-one with a point in the world coordinates. However, an event represents the change in brightness of a pixel between time $t$ and time $t - \Delta t$, which corresponds to two adjacent positions in the world space (shown in Figure 4). Due to the different imaging principles of images and events, warping methods designed for images cannot be directly applied to events. For event-based warping, when projecting

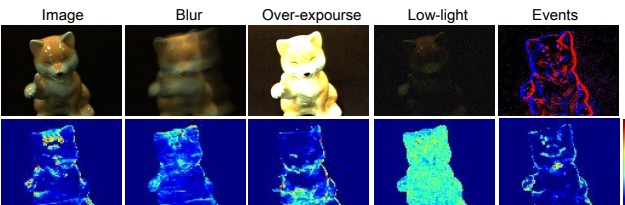

Figure 3: Illustration of specular reflection confidence in different scenes: (Top). Visualization of images / events from different scenes; (Bottom). The corresponding specular reflection confidence.

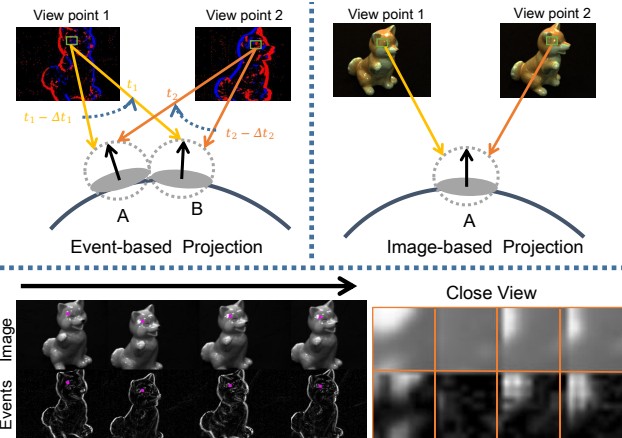

Figure 4: (Top). Illustration of the differences between event-based projection and image-based projection: For events, for two small adjacent areas A and B on the surface, the projection of A and B on the event camera is a small pixel patch over the time interval from $t - \Delta t$ to $t$. In contrast, for images, for a small area A on the surface, the projection of A on the image is a small pixel patch; (Bottom). The correspondence of the same world space location on event slices and images from different viewpoints.

events from different viewpoints onto the same reference view, it is necessary to determine whether the two points in world space corresponding to the events from the source view are the same as the events in the reference view.

Because events are asynchronous and sparse, to reduce computational load, we slice the events along the temporal dimension, with events within the same slice sharing the same time. Since a single event carries too little information, to increase the effective information, we perform warping on pixel patches rather than on single pixel points.

Given a reference view event slice $E_{ref}$ and a series of source view event slices $\{E_{src}^i\}$, our objective is to warp the source view event slices to the reference view. Shown in Figure 4, each event slice is associated with two positions in the world space, which we denote as $\mathbf{p}_{\{A,B\}}$. And these two positions have corresponding normal $\mathbf{n}_{\{A,B\}}$, rotation matrix $\mathbf{R}_{\{A,B\}}$, and translation vector $\mathbf{t}_{\{A,B\}}$. Like [15], the point $x$ in the pixel patch $q_{ref}$ of reference event slice $E_{ref}$ is related to the corresponding point $x'$ in the pixel patch $q_{src}^i$ of the source event slice $E_{src}^i$ via the plane-induced homography $H$ [25]:

$$x = \mathbf{H}_A x', x = \mathbf{H}_B x', \quad (4)$$

$$\mathbf{H}_i = \mathbf{K}\left(\mathbf{R}_i^{src}(\mathbf{R}_i^{ref})^T + \frac{\mathbf{R}_i^{src}((\mathbf{R}_i^{src})^T t_i^{src} - (\mathbf{R}_i^{ref})^T t_i^{ref})\mathbf{n}_i^T}{\mathbf{n}_i^T \mathbf{p}_i}\right)\mathbf{K}^{-1}, \quad (5)$$

where $i \in \{A, B\}$, $\mathbf{K}$ donates the internal calibration matrix. We determine the validity of the source view event patch by calculating the distance between the center positions of the projections of two patches:

$$|\mathbf{H}_A x_{center} - \mathbf{H}_B x_{center}| < \frac{s}{2}, \quad (6)$$

where we set patch size $s$ to 11. Figure 4 illustrates the correspondence of the same positions in world space as viewed from different perspectives.

***Measuring the similarity between event patches.*** When measuring the similarity between image patches, previous works [15] convert RGB images to grayscale images and then use Normalized Cross Correlation (NCC) to assess the similarity between two grayscale image patches. However, since events are sparse, discrete values with polarity (+1, -1), and subject to noise, directly applying

NCC to event patches may not be effective. To address this, we propose to accumulate events within a certain time interval and normalize the accumulated event patch. The specific steps are as follows:

$$E = |\frac{\sum_{\Delta t} e}{k}|, \quad (7)$$

where $\sum_{\Delta t} e$ represents the accumulated events within the specified time interval $\Delta t$, $k$ is an empirically determined constant and we set $k$ to 10. To mitigate the impact of noise in events on similarity measurement, we incorporate noise modeling when computing the NCC.

$$NCC(x,y) = \frac{Cov(x + \mathcal{N}(\mu,\sigma), y + \mathcal{N}(\mu,\sigma))}{\sqrt{Var(x + \mathcal{N}(\mu,\sigma)) \cdot Var(y + \mathcal{N}(\mu,\sigma))}}, \quad (8)$$

where $\mathcal{N}$ is a normal distribution with a mean $\mu$ of 0 and a standard deviation $\sigma$ of 0.01.

***Events clues.*** Based on the event-based warping and the metric for measuring similarity, we can obtain the similarity between the event patch of the reference view and a series of event patches from source views. Similar to [20], we believe that only the local region of the partial event patch contains specular reflections. Therefore, we use the variance of these similarities as the confidence measure for specular reflection,

$$M_s = Var(\{1 - NCC(E_{ref}, E_{src}^i)\}), \quad (9)$$

The specular reflection confidence obtained from the events can serve as a clue for specular highlights, which helps mitigate the impact of specular reflections on intrinsic decomposition. Specifically, this confidence can be used as a soft mask, applied to the loss function to reduce the influence of specular regions. Compared to

image-based approaches, event-based methods demonstrate stable performance even in extreme scenarios (shown in Figure 3).

## 3.3 Event-based Intrinsic Decomposition

The overview of our method is illustrated in Figure 5. We parameterize the 3D scene into four fields: the scene geometry is described by a signed distance field $\mathcal{G}$; surface normal is described by a normal field $\mathcal{N}$; material properties of the surface are described by a BRDF field $\mathcal{B}$; and the lighting of the scene is described by a light field $\mathcal{S}$. We optimize these four fields using both events and images.

**Scene geometry.** Following [78, 81], we represent the scene geometry as a Signed Distance Field (SDF). We obtain the position $\mathbf{x}$ of the object's surface points from this field, as well as the corresponding normal $\mathbf{n}_{\text{sdf}}$ and radiance $\mathbf{R}_{\text{vol}}$.

$$\mathcal{G} : \mathbf{x} \rightarrow \{\mathbf{d}, \mathbf{n}_{\text{sdf}}, \mathbf{R}_{\text{vol}}\}, \tag{10}$$

where $\mathbf{d}$ represents the signed distance from the point $\mathbf{x}$ to the closest surface. Similar to [66], this field is optimized using a volume rendering technique. Due to the poor image quality in extreme scenarios, events tend to yield better imaging results compared to traditional images. Similar to [57], we utilize events as a supervisory signal to guide the optimization process, ensuring that the background color in the volume rendering closely matches the background color observed in the images. The loss $\mathcal{L}_{\text{vol}}$ can be written as,

$$\mathcal{L}_{\text{vol}} = ||M_{\text{color}}\Theta \sum_{\Delta t} e_{\mathbf{u}} - M_{\text{color}}E_{\mathbf{u}}||_2, \tag{11}$$

$$E_{\mathbf{u}} = \ln(\mathbf{R}_{\text{vol}}(\mathbf{u}, \text{t})) - \ln(\mathbf{R}_{\text{vol}}(\mathbf{u}, \text{t} - \Delta \text{t})), \tag{12}$$

where $\mathbf{R}_{\text{vol}}$ is the color output of volume rendering[3], $M_{\text{color}}$ is described in [57], which is used to map the color information of the events onto the three RGB channels.

**Surface normals of the scene.** The surface normals within a 3D scene can be formulated as a normal field, which is encoded by a multi-layer perceptron (MLP). This MLP takes a point location $\mathbf{x}$ as input and outputs a surface normal $\mathbf{n}$.

$$\mathcal{N} : \mathbf{x} \rightarrow \mathbf{n}, \tag{13}$$

We use the normal $\mathbf{n}_{\text{vol}}$ output by $\mathcal{G}$ as the supervisory signal and loss $\mathcal{L}_{\text{normal}}$ can be written as,

$$\mathcal{L}_{\text{normal}} = ||M_s \mathbf{n} - M_s \mathbf{n}_{\text{sdf}}||_2, \tag{14}$$

where $M_s$ is described by Equation (9).

**Material properties of the surface.** Following [75, 78], the material properties of the surface can be formulated as a BRDF field, which is encoded by a MLP. This MLP takes a point location $\mathbf{x}$ as input and outputs a base color $\mathbf{b}$, a roughness $r$ and a metallic $m$.

$$\mathcal{B} : \mathbf{x} \rightarrow \{\mathbf{b}, r, m\}, \tag{15}$$

We use a simplified Disney principled BRDF [8] parameterization. The BRDF $f$ in Equation (3) can be computed as,

$$f(\omega_o, \omega_i, \mathbf{n}) = f_d + f_s(\omega_o, \omega_i, \mathbf{n}), \tag{16}$$

$$f_d = \frac{1 - m}{\pi} \cdot \mathbf{b}, \tag{17}$$

---

[3]More details are provided in the supplementary material.

$$f_s(\omega_o, \omega_i, \mathbf{n}) = \frac{D(h; r) \cdot F(\omega_o, h; \mathbf{b}, m) \cdot G(\omega_i, \omega_o, h; r)}{4 \cdot (\mathbf{n} \cdot \omega_i) \cdot (\mathbf{n} \cdot \omega_o)}, \tag{18}$$

where $f_d$ is diffuse term, $f_s$ is specular term, $h$ is the half vector between the incident direction $\omega_i$ and the viewing direction $\omega_o$. D, F, and G refer to the normal distribution function, the Fresnel term, and the geometry term respectively[4]. The loss $\mathcal{L}_{\text{base\_color}}$ can be written as,

$$\mathcal{L}_{\text{base\_color}} = ||M_s \frac{\mathbf{b}}{||\mathbf{b}||_2} - M_s \frac{\mathbf{R}_{\text{vol}}}{||\mathbf{R}_{\text{vol}}||_2}||_2, \tag{19}$$

**Lighting of the scene.** Like [75, 78], the incoming lights in the scene can be formulated as a neural incident light field, which is recorded by a MLP. The MLP takes a point location $\mathbf{x}$ and a direction $\omega_i$ as inputs, and returns an incident light $\mathbf{L}_i$,

$$\mathcal{S} : \{\mathbf{x}, \omega_i\} \rightarrow \mathbf{L}_i, \tag{20}$$

**Event-based surface rendering equation.** Due to events being sparse and asynchronous, and containing noise, to improve efficiency and reduce the impact of noise, we employ the same slicing method as in section 3.2 for surface rendering. The surface rendering equation for events established in section 3.1 can be reformulated as follows,

$$E_{\mathbf{u}} = \ln(\mathbf{I}(\mathbf{u}, \text{t})) - \ln(\mathbf{I}(\mathbf{u}, \text{t} - \Delta \text{t})), \tag{21}$$

where $\mathbf{I}(\mathbf{u}, t)$ is described by Equation (3). And the $E_{\mathbf{u}}$ can be supervisd by events generated at $\mathbf{u}$, the loss $\mathcal{L}_{\text{surf}}$ is,

$$\mathcal{L}_{\text{surf}} = ||M_{\text{color}}E_{\mathbf{u}} - M_{\text{color}}\Theta \sum_{\Delta t} e_{\mathbf{u}}||_1. \tag{22}$$

Since Equation (21) represents a differential relationship, we use the radiance $\mathbf{R}_{\text{vol}}$ output from the first stage as an additional supervisory signal to constrain radiance $\mathbf{I}(\mathbf{u}, t)$.

$$\mathcal{L}_{\text{surf\_vol}} = ||\mathbf{I}(\mathbf{u}, t) - \mathbf{R}_{\text{vol}}(\mathbf{u}, t)||_1. \tag{23}$$

**Optimization strategy.** Like [78], our strategy is divided into two distinct optimization stages for those fields. Initially, we employ a combination of events and images to optimize the scene's geometric field $\mathcal{G}$. Following this, we enter the second stage, wherein $\mathcal{G}$ is freezed, allowing us to focus on the simultaneous optimization of the normal field $\mathcal{N}$, the BRDF field $\mathcal{B}$, and the light field $\mathcal{S}$.

## 4 EXPERIMENTS

In this section, we evaluate our proposed method using both synthetic and real-world datasets. We conduct ablation studies to validate the effectiveness of each module.

## 4.1 Datasets and Metric

**Synthetic data.** To compensate for the absence of a dataset comprising events, images (including overexposed, blurred, and low-light), base color GT, and normal GT. We augment the synthetic dataset from NeRF synthetic dataset [47] by extending it with seven synthetic scenes, *i.e.*, chair, drums, ficus, hotdog, lego, materials, and mic. First, we utilize Blender [50] to synthesize 1000 continuous HDR images, and composite events using the same method as [74].

---

[4]We adopt similar implementation of D, F, and G as in previous works [75, 78] and details are provided in the supplementary material.

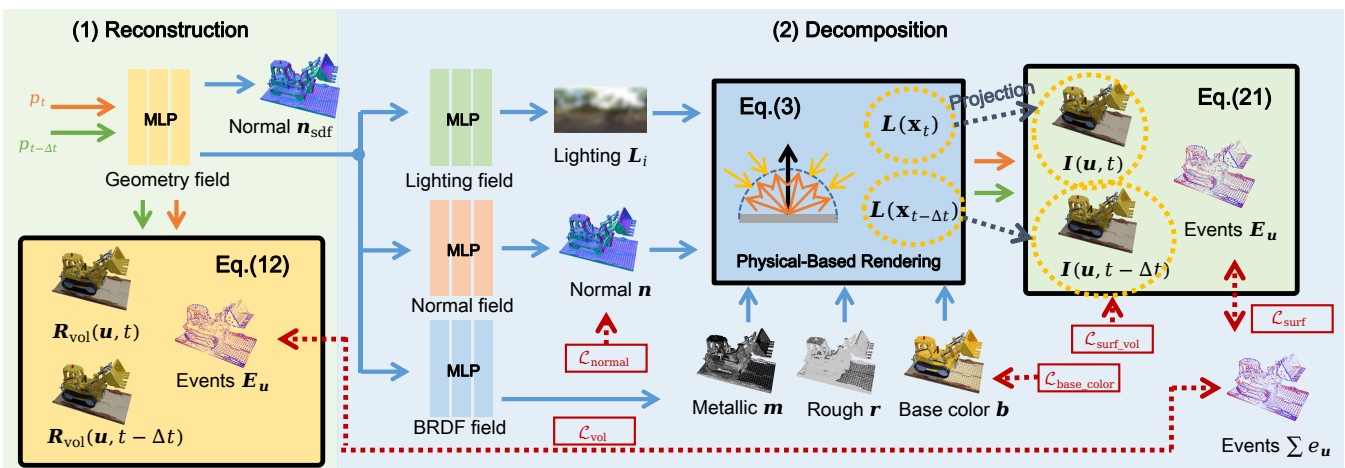

Figure 5: The overview of our method. The first part uses events and images to reconstruct a geometry field $\mathcal{G}$. The second part involves freezing the geometry field and using information from the geometry field along with events to reconstruct three fields: Lighting field $\mathcal{S}$, Nomral field $\mathcal{N}$ and BRDF field $\mathcal{B}$.

We uniformly sample 100 HDR images and adjust the exposure via method [30] to synthesize overexposed images. By employing method [40], we reduce the exposure and add noise to the images to synthesize underexposed images (low-light). Using method [54], we synthesized blurred images.

In addition to the aforementioned training data, we have also synthesized 200 testing images from different viewpoints than the training data, along with the corresponding ground truth for base color and normals. In summary, our synthetic dataset contains images under four different conditions, as well as the corresponding events. We will evaluate our results and SOTA methods on these three datasets.

**Real-world data.** To test our approach's robustness, we collected real data using the DAVIS 346C sensor [57] from three scenes, capturing 16-second monocular videos (each video is about 16 seconds and 500 frames) with events and frames. Frame poses were estimated using COLMAP [60]. Although the poses may not be perfectly accurate due to potential blurring, they were sufficiently estimated for our purposes. We uniformly selected 100 images to serve as the training data. Additionally, we randomly selected 100 images for testing.

**Metric.** Our evaluation encompasses normal quality (measured by MAE [9]), novel view synthesis, and base color (measured by PSNR [28], SSIM [28], and LPIPS [82])

### 4.2 Comparison with State-of-the-Art Methods

Due to significant differences in the scenes compared to those used in previous methods, some methods did not work in our scenarios. In the main text, we primarily focus on comparing state-of-the-art (SOTA) neural field-based intrinsic decomposition methods GD-IR [41] and its integration with other methods. The results of other methods will be presented in the supplementary materials.

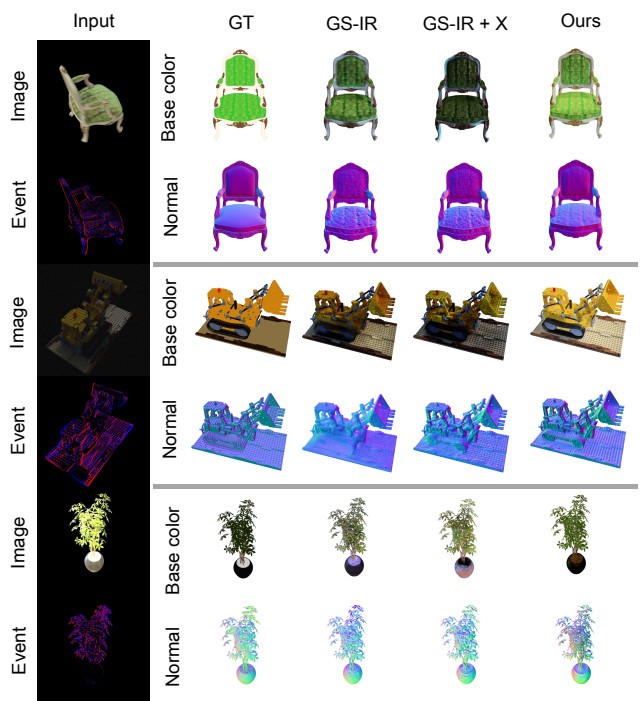

Figure 6: Qualitative comparison on our synthetic dataset. We visualize the estimated normal, base color results of our method and SOTA methods on three scenes (From top to bottom, they are blurred, and low-light, and overexposed respectively). In these three scenarios, X represents methods WZ23 [69] , WY21 [67] , and YH23 [74] respectively.

***The results of the SOTA methods.*** Table 1 provides a quantitative comparison of synthetic data, highlighting that our approach maintains stable performance even under conditions of input image degradation. This resilience is attributed to the high temporal resolution and dynamic range benefits afforded by events.

Figure 6 displays the visualization results under three different types of image degradation. The degradation of images leads to inconsistencies in multi-view information, making it challenging for the network to extract useful information. However, our method circumvents the issues associated with image degradation by extracting scene information directly from events, thereby avoiding the pitfalls of degraded image quality.

***The results of the combination of SOTA methods.*** To ensure a fair comparison given that our method simultaneously inputs events and images, we selected three event-guided methods to enhance images degraded by different factors: YH23 [74] for correcting overexposure [5], WZ23 [69] for deblurring, and WY21 [67] for enhancing low-light images. This approach allows us to evaluate the effectiveness of our framework against specialized methods tailored to address specific types of image degradation. Additionally, we compared methods that combine event-based NeRF method LL23 [42] with GS-IR [41][6]. These approaches are similar to ours, as they both extract valuable information from events and consider the consistency across multiple viewpoints.

Table 1 displays the quantitative results of the combined methods. It becomes clear that merely combining events with images straightforwardly fails to capitalize on the unique advantages of events effectively. This inefficiency stems from the event-enhanced method of applying modifications on a frame-by-frame basis, which leads to the loss of multi-view consistency in the enhanced images. Additionally, the methods that combine event-based NeRF with GS-IR [41] perform better than those that simply use events to enhance images. This improvement is due to event-based NeRF's ability to extract scene information from events while maintaining multi-view consistency. However, this approach only utilizes event information during the reconstruction of the 3D scene and does not fully exploit it during the decomposition phase. In contrast, our method makes full use of event information in both the 3D scene reconstruction and decomposition stages, resulting in superior outcomes.

Figure 7 showcases the visual results, clearly illustrating that our method achieves the best performance.

***The results of real-world data.*** Figure 8 presents the results on real-world data which, unlike synthetic data, are constrained by device capabilities, resulting in images with a resolution of only 346x260 and significantly poorer image quality. Despite the low quality of the images, our results still outperform comparative methods. For the results of the remaining real-world data, please refer to the supplementary materials.

## 4.3 Ablation Studies

To validate the effectiveness and necessity of the components of our method, we compare it with its two variants:

[5]We use HDR images as the input for the intrinsic decomposition method.
[6]The images output by LL23 [42] are used as input for GS-IR [41].

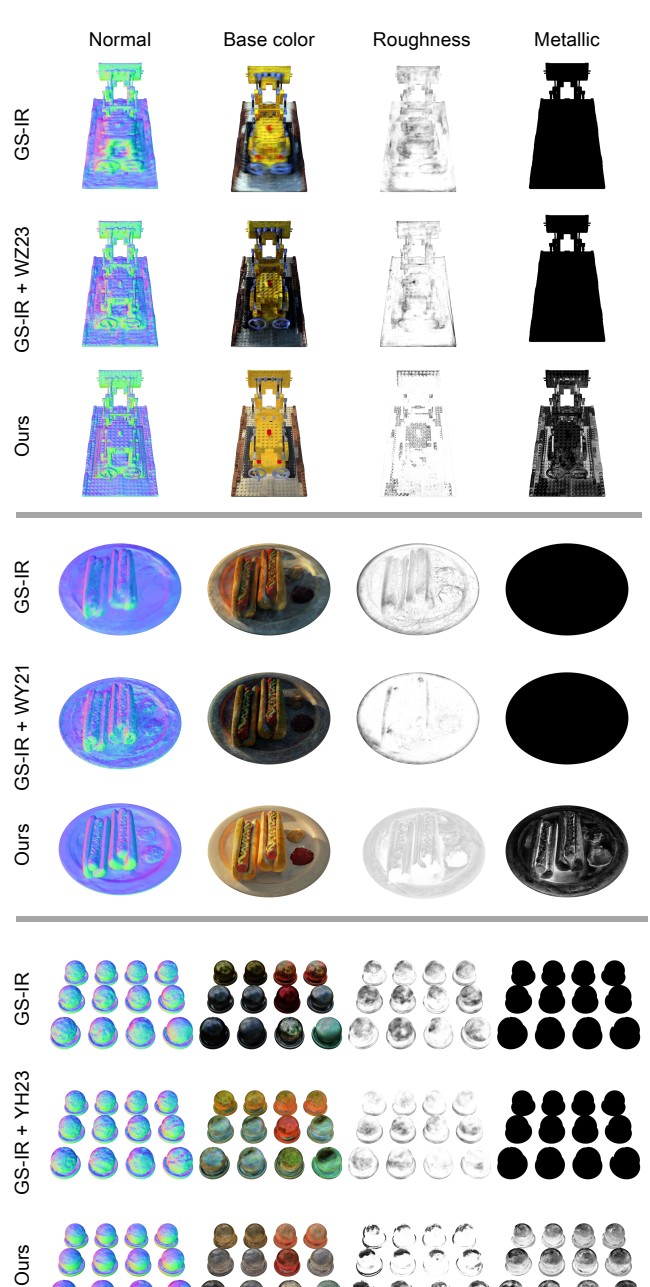

**Figure 7: Qualitative comparison on synthetic dataset. We visualize the estimated normal, base color, roughness and metallic of our Event-ID and baseline methods. (From top to bottom: scenes with blur, low light, and overexposure.)**

- **Variant A:** Without event surface rendering loss $\mathcal{L}_{\text{surf}}$.
- **Variant B:** Without specular-related cues $M_s$.

Table 2 presents the quantitative results for different variants. It is clear that removing any component of our method results in decreased performance.

**Table 1: Quantitative comparison on synthetic dataset.**

| Conditions | Methods | Normal | Base Color | | | Novel View | | |
|---|---|---|---|---|---|---|---|---|
| | | MAE(°) ↓ | PSNR ↑ | SSIM ↑ | LPIPS ↓ | PSNR ↑ | SSIM ↑ | LPIPS ↓ |
| Over-exposure | GS-IR [41] | 28.1769 | 15.3427 | 0.8498 | 0.1675 | 18.5523 | 0.9156 | 0.0901 |
| | GS-IR [41]+YH23 [74] | 28.2794 | 15.7839 | 0.8558 | 0.1594 | 15.1489 | 0.8822 | 0.1259 |
| Low light | GS-IR [41] | 44.7076 | 16.2018 | 0.8461 | 0.1745 | 16.3147 | 0.8875 | 0.1120 |
| | GS-IR [41]+WY21 [67] | 33.0106 | 15.3934 | 0.8273 | 0.1748 | 23.4842 | 0.8987 | 0.1066 |
| Blur | GS-IR [41] | 35.5682 | 15.6107 | 0.8588 | 0.1832 | 22.6323 | 0.8916 | 0.1196 |
| | GS-IR [41]+WZ23 [69] | 28.9436 | 15.5308 | 0.8452 | 0.1727 | 23.5555 | 0.9026 | 0.0943 |
| | GS-IR [41]+LL23 [42] | 27.3428 | 17.1431 | 0.8403 | 0.1694 | 22.1696 | 0.8993 | 0.0899 |
| | Ours | **20.1804** | **20.4238** | **0.8930** | **0.1140** | **25.3671** | **0.9222** | **0.0726** |

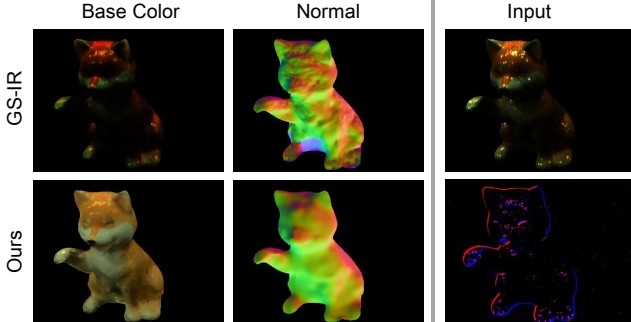

Figure 8: Qualitative comparison on real-world dataset in low-light scene. We visualize the estimated based color, normal, and input of our Event-ID and GS-IR [41].

**Table 2: Quantitative comparison of ablation study on synthetic dataset.**

| Variants | Normal | Base Color | | | Novel View | | |
|---|---|---|---|---|---|---|---|
| | MAE(°) ↓ | PSNR ↑ | SSIM ↑ | LPIPS ↓ | PSNR ↑ | SSIM ↑ | LPIPS ↓ |
| w/o $\mathcal{L}_{surf}$ | 26.0321 | 18.4492 | 0.8669 | 0.1261 | 23.9775 | 0.9076 | 0.0921 |
| w/o $M_s$ | 23.7485 | 19.0964 | 0.8653 | 0.1194 | 24.8082 | 0.9147 | 0.0720 |
| Ours | **20.1804** | **20.4238** | **0.8930** | **0.1140** | **25.3671** | **0.9222** | **0.0726** |

***Effectiveness of the clues from events.*** To verify the effectiveness of the specular-related cues obtained from events, we remove the $M_s$ from the loss $\mathcal{L}_{normal}$ and $\mathcal{L}_{base\_color}$ as a part of our comparative experiment. Figure 9 and Figure 10 respectively show the results on base color estimation and normal estimation. It can be observed that utilizing specular-related clues to separate specular highlights is beneficial for the reconstruction of base color and normals in specular regions. The reason is that for base color, the specular regions reflect the color of the light source rather than the color of the object itself. Masking out the pixels in the specular positions is tantamount to reducing the input of incorrect information, allowing the network to be undisturbed by specular information. As for normal estimation, the normal is constrained by the output of the first part, and the normal output from the first stage is inaccurate in the specular areas due to not considering the effect of specular reflections. Lowering the weight of the normals in these regions also equates to reducing the input of incorrect information. Therefore, in the experimental results with $M_s$, the highlighted box areas exhibit a relatively smooth surface of normal.

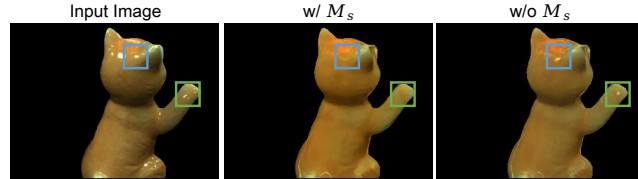

Figure 9: Visualization of ablation study on the effectiveness of specular-related clues ($M_s$) to base color estimation. The placement of the boxes highlights the differences between the two outcomes.

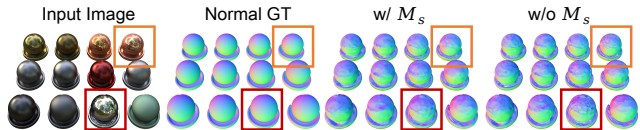

Figure 10: Visualization of ablation study on the effectiveness of specular-related clues ($M_s$) to normal estimation. The placement of the boxes highlights the differences between the two outcomes.

## 5 CONCLUSION & LIMITATIONS

In conclusion, our research presents a intrinsic decomposition framework that effectively harnesses event-based data to overcome the limitations of conventional imaging under adverse conditions. By integrating an event-based reflectance model with specular-related event clues, we have enabled robust 3D scene reconstruction and intrinsic decomposition in challenging scenarios such as low-light, blur, and overexposure. Our method not only maintains consistency across multiple viewpoints but also improves upon the reliability of intrinsic decomposition, paving the way for practical applications in fields requiring precise scene interpretation and manipulation. This work signifies a considerable advancement in the utilization of event cameras for complex imaging tasks and sets the stage for future explorations in the domain.

***Limitations.*** Our method does not model shadows, and the current resolution of event cameras limits the application of this technique. However, these issues do not hinder the exploration of this scientific question, and we believe that with advancements in event camera technology and its commercial development, the resolution and imaging quality of event cameras will improve.

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
