# OpenReview forum: "Event-ID: Intrinsic Decomposition Using an Event Camera"
_acmmm.org/ACMMM/2024/Conference — MM2024 Poster_

### Official Review · Reviewer_JWAg · 2024-05-20

**Rating:** 3
**Confidence:** 3

**Summary:**

This article proposes an object reconstruction method based on event clues and image input, which has achieved success in challenging data (blurring, exposure, etc.). The core of the article is the introduction of an event guidance module, which establishes the relationship between event pixel speed and 3D motion based on event warping.  Then, multiple MLP models were used to establish a scene decomposition model, and projection was performed based on volume rendering. The method achieved high-quality results in complex scenes by using event clues to guide decomposing scenes.

**Strengths:**

(1) Compared to previous methods, the proposed method utilizes event clues to enhance the consistency of specular regions.
(2) Due to the high dynamic range and resolution of event clues, excellent imaging quality has been achieved in complex scenes.

**Limitations:**

(1) Line 121-124, how to understand events to mitigate the impact of specular reflection on the intrinsic decomposition tasks (normal, material, lighting, etc). Although relevant explanations were made in section 3.3,  event cues are used in sufface normal and material properties term. But how did the event clues affect the geometry and lighting term?

(2) The division of diffuse and specular regions in Section 3.2 is unclear that the confidence evaluation threshold for $M_s$ is not clearly defined. The experiment lacks detailed descriptions and is difficult to reproduce. Or will it be open source code?

(3) The qualitative comparison results lack a GT perspective and cannot be compared intuitively (Fig 7,8). Fig 6 GT appears to be a processed image, and there is no comparability between Ours and GT

(4) Figure 9 shows that w/o $M_s$ achieved better mirror effect, does this mean that specular cues are invalid?

(5) The comparison methods are not enough. They only combine GS-IR and different data processing methods (e.g. LL23, WZ23 for different scenarios), and are not compared to the most relevant methods in this article, such as [75, 78] similar decomposition models and [66, 57] scene geometry reconstruction models. Unable to make accurate evaluations from existing comparative methods.

(6) The ablation experiment is not sufficient, and it seems that no ablation experiments have been conducted on the event monitoring signals $L_{surf-vol}$ and $L_ {vol}$.


The method logic should be correct and effective, but more comparative experimental should be provided.

**Suitability:**

2

---

### Official Review · Reviewer_vF2q · 2024-05-24

**Rating:** 6
**Confidence:** 3

**Summary:**

This paper is about to use event camera for intrinsic decomposition.

**Strengths:**

This paper proposed a novel event-based intrinsic decomposition framework that can estimate the geometry, materials and lighting simultaneously. They established an event-based reflectance model and introduced an event-based warping method to extract specular clues. The proposed method can not only maintain the consistency across multiple viewpoints but also improve upon the reliability of intrinsic decomposition. Both qualitative and quantitative comparisons demonstrate its superiorities, especially under low-light, blur and over-exposure situations.

**Limitations:**

No other limitations of this study were identified in addition to the limitations mentioned by the authors themselves.

**Suitability:**

3

---

### Official Review · Reviewer_2KUe · 2024-05-26

**Rating:** 2
**Confidence:** 4

**Summary:**

This work proposes an intrinsic decomposition framework that decomposes RGB  into base color, normal and BRDF (roughness, metallic).  To achieve this, they utilize event cameras, which can measure subtle changes in brightness at a very high frame rate. To summarize, they utilize an event-based reflectance model for intrinsic decomposition.

**Strengths:**

- A major technical novelty in the paper is using an event camera for intrinsic decomposition.
- They augmented the synthetic dataset to create an event camera setup under different lighting conditions.

**Limitations:**

- **Missing details on resolution for the event camera**: The spatial resolution of the event camera is low compared to RGB cameras[R1]. However,  for the task of novel-view synthesis, we expect high-quality inputs at high resolution. This has not been discussed anywhere in the proposed work addressing this challenge for real-world capture.

- **Missing comparison with some baselines for intrinsic decomposition.**: The authors have missed out comparison with works like Intrinsic-NeRF[R2],  NerFactor[R3] which also solves intrinsic decomposition task.

- **Missing qualitative results on consistent "base color"**: For intrinsic decomposition, it is very necessary that the estimated "base color" is consistent across all views. But in the "potted plant" scene in the results shown in the supplementary video (timestamp 01:01 to 01:07), we can see a color change in the plant's leaf. Similarly, there is a slight color change in the drum as well.


[R1]  Han, Jin, Yixin Yang, Chu Zhou, Chao Xu, and Boxin Shi. "Evintsr-net: Event guided multiple latent frames reconstruction and super-resolution." In Proceedings of the IEEE/CVF International Conference on Computer Vision, pp. 4882-4891. 2021.

[R2] Ye, Weicai, Shuo Chen, Chong Bao, Hujun Bao, Marc Pollefeys, Zhaopeng Cui, and Guofeng Zhang. "Intrinsicnerf: Learning intrinsic neural radiance fields for editable novel view synthesis." In Proceedings of the IEEE/CVF International Conference on Computer Vision, pp. 339-351. 2023.

[R3] Zhang, Xiuming, Pratul P. Srinivasan, Boyang Deng, Paul Debevec, William T. Freeman, and Jonathan T. Barron. "Nerfactor: Neural factorization of shape and reflectance under an unknown illumination." ACM Transactions on Graphics (ToG) 40, no. 6 (2021): 1-18.

**Suitability:**

3

---

### Meta-Review · Area_Chair_z9sa · 2024-07-05

**Recommendation:** Accept (Poster)
**Confidence:** 5

**Metareview:**

This paper received mixed scores in the first round of review. Thanks to the rebuttal, reviewers believe that the rebuttal cleared most of their concerns, and all reviewers are on the positive side after rebuttal. AC believes that this paper should be communicated, and requires the authors to carefully prepare the camera ready, as they promised in the rebuttal.